# The Right to Development and disparities in healthcare access: Qualitative study from rural Ontario, Canada

Amal Jawad[1]*, Bonny Ibhawoh[2], Lisa Schwartz[3], Andrew Kapoor[4]

1 Mary Heersink School of Global Health, Department of Health Science, McMaster University, Hamilton, Ontario, Canada, 2 Department of History, McMaster University, Hamilton, Ontario, Canada, 3 Department of Health Sciences, McMaster University, Hamilton, Ontario, Canada, 4 Department of Medicine, McMaster University, Hamilton, Ontario, Canada

* jawada4@mcmaster.ca

## Abstract

The Right to Development is the UN-recognised principle that frames development as a human right, emphasising equitable access to resources and opportunities. As enshrined in the UN Declaration on Human Rights, it affirms that all individuals should be able to participate in, contribute to, and benefit from development processes, including healthcare. This study examines how ethnic minorities in Oxford County, rural Ontario, perceive their healthcare access and identify barriers to their right to development. Using a qualitative interpretive approach, in-depth semi-structured interviews were conducted with 25 participants, 16 men and 9 women, between January and April 2025. Data were analysed through Braun and Clarke's six-step method for thematic analysis, revealing two central themes: constraints on social support and equity and inclusion policy. The first theme highlights compounded challenges, including ethnoracial stratification, social isolation, and limited social support. The second theme reflects participants' views on accessibility, resource allocation, and inclusivity of healthcare, pointing to the need for targeted policy interventions. Findings demonstrate that while participation is critical, meaningful healthcare equity requires more substantial commitments to universality and accessibility. Applying the right to development framework advances understanding of healthcare disparities by linking inclusive participation with the practical delivery of equitable services. This study carries significant policy implications, underscoring the importance of addressing structural and cultural barriers, strengthening community support systems, and ensuring fair distribution of healthcare resources. Situating the rural Ontario case within broader global debates, the research offers insights applicable to rural and underserved populations worldwide.

**Data availability statement:** This research relies on primary data gathered and analysed as part of a field study. Access to these data is restricted under the terms of the Hamilton Ethics Board approval, which prohibits public release or external sharing of participant information. The data underlying this study cannot be made publicly available due to ethical restrictions imposed by the Hamilton Integrated Research Ethics Board (HiREB). Although the dataset has been de-identified, it contains sensitive qualitative information that could pose a risk of indirect identification of participants if shared publicly. These restrictions were imposed as part of the ethics approval process in order to protect participant confidentiality and comply with institutional research ethics requirements. Requests for access to the data may be considered on a case-by-case basis, subject to approval by the Hamilton Hamilton Integrated Research Ethics Board (HiREB) 237 Barton Street Hamilton, Ontario, L8L 2X2 Email: eREBHelpdesk@hhsc.ca / lukac@hhsc.ca Phone: 905 521-2100 Ext 70014 Website: [www.HiREB.ca]www.HiREB.ca.

**Funding:** The author(s) received no specific funding for this work.

## Introduction

Ethnic minority communities are fundamental to Canada's social and economic fabric. In the Canadian context, an *ethnic group* refers to a population segment that shares common ancestral origins, cultural heritage, language, or traditions, including visible minorities, immigrants, refugees, and newcomers [1, 2]. Those groups face significant disparities.

Health disparities remain a significant public health and social justice issue even in wealthy nations [3], often linked to social, economic, and environmental disadvantages [4]. In Canada, these disparities are acute in rural and remote regions due to limited healthcare access, geographic isolation, and cultural barriers [5,6]. They stem from structural inequalities in the distribution of resources, wealth, and power [7,8], disproportionately affecting ethnic minorities [9].

Canada is home to over 450 ethnic groups [1]. Boyd classifies at least 11 of these populations as "visible minorities," including non-Caucasian or non-white individuals, immigrants, refugees, and newcomers [2]. Those communities are integral to Canada's social and economic fabric, contributing substantially to the nation's growth through diverse channels [10,11]. For instance, in 2022, international students, many of whom belong to these communities and their visiting relatives and friends, generated $37.3 billion in economic activity across Canada [12]. This spending added $30.9 billion to Canada's GDP, accounting for 1.2% of the national total. Ontario, home to the largest number of international students, contributed the most at $16.9 billion, accounting for 54.6% of the total GDP impact, followed by British Columbia at 18.4% and Quebec at 12.4% [13]. These figures highlight the essential role that international students and their communities play in driving Canada's economic prosperity [10]. Migrant groups play a vital role in supporting rural economies, bringing diverse skills, launching businesses, and addressing labour shortages [11,14]. However, despite their significant contributions, ethnic minorities remain particularly vulnerable at the intersection of rural and migration-related disadvantages.

As a result, even within Canada's universal healthcare system [15], the progressive realisation of the right to health, that is, the principle under international human rights law that governments must take continuous and deliberate steps to improve access to healthcare over time, within their available resources, is not achieved equitably for all populations [16]. Yet, these groups face compounded barriers from both migration-related and rural disadvantages, limiting equitable realisation of the right to health [15]. In rural Ontario, ethnic minorities encounter persistent barriers such as language challenges, lack of trust in institutions, transportation issues, cultural differences, and unfamiliarity with the healthcare system [17]. Two interlinked challenges, widespread shortages of healthcare resources and social exclusion, further restrict their participation in health service planning [18]. These issues are urgent given that over 1,500 primary care positions remain unfilled and 60% of rural municipalities lack mental health services [19].

Addressing disparities requires public health policies rooted in human rights and equity [20]. The International Covenant on Economic, Social and Cultural Rights (ICESCR), adopted by the United Nations General Assembly in 1966 and

entering into force in 1976, enshrines both material needs, such as food, health, shelter, education, and work and vital non-material aspects such as participation and freedom [21,22]. Notably, the ICESCR also establishes the principle of "progressive realisation," meaning that states must take continuous and deliberate steps, within their available resources, to progressively realise these rights [23]. Within international human rights policy, the principle of progressive realisation, articulated in Article 2(1) of the ICESCR, acknowledges that economic, social, and cultural rights may not be achieved immediately by all states. Instead, governments are required to take deliberate and continuous steps, within their available resources, to progressively ensure the fulfilment of rights such as the right to health, education, housing, and adequate food. John Rawls' philosophical framework emphasises that everyone should have access to "basic social goods," a principle that aligns with the RtD's commitment to equitable access to resources and opportunities [24]. Extending this logic, the Right to Development was articulated as "an inalienable human right by virtue of which every human person and all peoples are entitled to participate in, contribute to, and enjoy economic, social, cultural and political development, in which all human rights and fundamental freedoms can be fully realised" [25]. The RtD encompasses development beyond material *basic needs* to embrace social, cultural, and political dimensions, promoting both collective and individual progress and emphasising ongoing improvements in the quality of life over time [26]. While Right to Development (RtD) discourse often focuses on the Global South, it has also been applied in developed contexts, including Indigenous participation in resource management in New Zealand [27] and efforts to address regional inequalities in the European Union [24]. The United States supported the RtD at the 1993 Vienna World Conference on Human Rights, endorsing paragraph 10 of the Vienna Declaration, which reaffirms the RtD as a universal, inalienable, and integral human right [28,29]. Canada was among the 171 states that adopted the Vienna Declaration and Programme of Action in 1993, which reaffirmed the RtD as a universal, inalienable right [29,30]. The present study adopts the RtD as its guiding framework to address health disparities in rural Oxford County, Ontario. The RtD is especially relevant in this context because it links health equity to broader issues of participation, resource distribution, and social inclusion dimensions often overlooked in affluent-country settings. By applying RtD, this study highlights not only gaps in access to healthcare but also structural inequities in development processes, offering insights that go beyond service delivery perspectives.

The study argues that integrating RtD principles into healthcare policy can ensure meaningful participation of ethnic minorities in decision-making while guaranteeing universality and accessibility of healthcare for all. Guided by this perspective, the central research question is: How do ethnic minorities perceive their healthcare access, and what barriers do they identify as most critical to their RtD in Oxford County, rural Ontario? Qualitative methods, including semi-structured interviews with ethnic minority groups, in-depth, open-ended questions, were conducted in Oxford County, rural Ontario. In this study, I first delve into the theoretical framework of RtD. The methodology section outlines the philosophical orientation, participation, data collection and data analysis. Findings from the lived experiences of ethnic minority residents identify key themes and barriers. The discussion, limitations, and future research consider how ethnic groups' perspectives can shape policy through both procedural and substantive values, ensuring that inclusion, universality, and accessibility are reflected in health policy and practice. In doing so, the study demonstrates how the RtD framework can serve as a practical tool for advancing equity in healthcare policy.

## Theoretical framework

The RtD, affirmed by the United Nations General Assembly in 1986, was shaped by long-standing debates about the indivisibility of civil, political, social, and cultural rights [31,32]. Its foundations lie in the Philadelphia Declaration of 1944, the United Nations Charter of 1945, and the Universal Declaration of Human Rights of 1948 [33]. The RtD defines development as a "comprehensive economic, social, cultural and political process," requiring active, free, and meaningful participation alongside fair distribution of benefits [25]. Conceived as a "vector" of human rights, it insists that the realisation of one right cannot justify setbacks in another [32,34].

National governments bear primary responsibility for ensuring the RtD, including equal access to resources, services, and opportunities and meaningful reforms to dismantle structural barriers [25], articles 2, 3, 8. This vision extends beyond material well-being to include education, health, housing, political voice, cultural inclusion, and ongoing improvements in living standards [26]. Global attention to the RtD has intensified since the establishment of an Intergovernmental Working Group in 2019, which drafted a Convention on the Right to Development presented to the Human Rights Council in 2021 [35]. If adopted, this Convention would elevate the RtD from a declaratory principle, as in the 1986 Declaration, to a legally binding obligation under international law. This global momentum highlights the importance of applying the RtD in local contexts, such as rural Ontario, where health disparities persist despite Canada's universal healthcare system.

The RtD also resonates with philosophical theories of justice. Rawls's difference principle allows inequalities only when they benefit the least advantaged [24,36]. Rawls argues that improvements in one group's well-being have value only if they are shared with those worse off, making justice contingent on raising the prospects of society's most marginalised members [24]. Rawls argues that social and economic inequalities are only justifiable if they produce real improvements for those who are least advantaged, as said "to the greatest advantage of the worst-off" or "raises the prospects of the least advantaged to the greatest possible degree" [36]. Liberal utilitarianism and basic needs approaches likewise emphasise guaranteeing essential goods, such as food, water, health, education, and shelter, to secure minimum living standards [24]. This limitation highlights a key critique of liberal theories, such as Dworkin's equality of resources, which protect individuals from bad luck without directly addressing accumulated advantage [37]. This limitation highlights a key critique of liberal theories, such as Dworkin's equality of resources, which protect individuals from bad luck without directly addressing accumulated advantage [37]. The RtD addresses this gap by insisting on not only formal equality of opportunity but also substantive measures to redress structural barriers and ensure that historically disadvantaged groups can participate in and benefit from development [31]. In contrast, Sen and Nussbaum's capabilities approach, which focuses on securing substantive freedoms for individuals to lead dignified lives, has informed the RtD's emphasis on participation and equitable benefit-sharing [25].

In contrast, Sen's and Nussbaum's capabilities approaches move beyond utilitarian and basic-needs frameworks by focusing on securing substantive freedoms and essential capabilities, such as the ability to access healthcare, education, and political participation, necessary for individuals and groups to lead dignified lives. In contrast, Sen's and Nussbaum's capabilities approaches move beyond utilitarian and basic-needs frameworks by focusing on securing substantive freedoms and essential capabilities, such as the ability to access healthcare, education, and political participation, necessary for individuals and groups to lead dignified lives [38].

The RtD bridges these frameworks by insisting that meeting basic needs is not enough; development must be equitable, participatory, and accountable [25]. Procedural values of transparency, inclusion, and meaningful public participation are inseparable from substantive outcomes such as reduced inequality and equitable resource distribution [39]. These principles are reflected in the Sustainable Development Goals [40], which frame inclusivity both as a strategy and an outcome of development. Scholars echo this emphasis on inclusion; for example, Ayeni et al. [41] argue that accessibility and participation are essential for a just society, while the European regional strategy identifies governance, policy coherence, and social inclusion as core pathways to sustainable development, even if implementation varies across national contexts [42].

Despite its wide acceptance, the RtD faces significant challenges. It lacks binding enforcement at the international level, leaving compliance dependent on state goodwill [43,44]. The definition of "development" remains broad and contested, complicating measurement and implementation. Tensions between state sovereignty and international oversight further limit progress, and without robust monitoring, participatory mechanisms, and resource allocation, RtD commitments risk remaining aspirational [43,44]. The pending Convention on the Right to Development may strengthen accountability. Still, its impact will remain constrained by structural limitations of the current UN system, including the non-binding nature of many enforcement mechanisms, reliance on state consent and political will, and limited capacity for compulsory compliance.

Canada's support for the RtD is reflected in its ratification of major human rights treaties such as the International Covenant on Economic, Social and Cultural Rights and the International Covenant on Civil and Political Rights [45]. Canadian officials played a central role in drafting the 1986 Declaration, and the RtD has influenced both federal and provincial human rights laws, including the Canadian Bill of Rights 1960 and the Charter of Rights and Freedoms 1982. In Ontario, the RtD directly informs the Ontario Human Rights Code [46]. The Vienna Declaration 1993 further emphasises that states are responsible for promoting and protecting all human rights, with follow-up resolutions urging national implementation [47,48].

However, Canada's commitment to social rights developed slowly. Before World War II, minimal state intervention and voluntarism dominated, and social rights, including the RtD, received little support [49]. Policy change was initiated in the mid-20th century, beginning with Ontario's first human rights law in 1944 and expanding through the consolidation of anti-discrimination legislation in 1962. The federal framework for multiculturalism, including the Multiculturalism Act 1988 and section 27 of the Charter, reflects Canada's ongoing commitment to equality, diversity, and inclusion [50]. Federal agencies such as the Department of Canadian Heritage and Immigration, Refugees and Citizenship Canada, continue to support these principles.

Nonetheless, as Satzewich and Liodakis note, gaps persist between legal protections and lived experiences [51]. Structural inequalities continue to affect access to healthcare and other services. This study treats equality as a guiding principle rather than a final destination. Beyond legislative recognition, meaningful equality requires ongoing policy action, public engagement, and evaluation. Equity-based approaches are essential in this process, as they address structural and contextual differences that prevent equal rights from producing equal outcomes. The RtD offers a valuable framework for guiding this process, emphasising participation, equitable resource distribution, and shared benefits from development. Realising this vision means creating policies that are not only inclusive in theory but effective in practice, particularly in sectors such as healthcare where disparities remain significant.

## Materials and methods

### Philosophical orientation

This qualitative research adopts a constructionist lens and uses semi-structured interviews to explore how participants construct and share meanings from their lived experiences. The study investigates how ethnic minorities in rural Oxford County, Ontario, experience and navigate barriers to healthcare services [52,53]. Guided by inductive reasoning, the study's qualitative research design was developed iteratively, with interview questions refined as the researcher actively identified and constructed themes through ongoing engagement with the data and participant perspectives [54]. The findings are presented through detailed, contextual narratives that centre participants' voices, rather than quantifying experiences. Fieldwork was conducted in natural settings, including community centres and participants' homes, to capture the dynamic and co-constructed nature of social reality. Reflexivity was integral throughout, as the researcher openly acknowledged their values, assumptions, and interpretive role, viewing knowledge creation as a collaborative process. Prioritising depth over breadth, the study's small-scale approach yields nuanced insights relevant to similar rural and minority populations. Integrating participants' lived realities, emerging theoretical ideas, and the researcher's positionality, this project offers an evolving, context-rich account of the barriers to healthcare access services.

### Recruitment and participants

The study's participants were members of ethnic minority groups. All participants resided in rural Oxford County, Ontario, were at least 18 years old, spoke English, and had regular interactions with healthcare services. Recruitment efforts employed multiple outreach strategies, including distributing digital flyers to local hospitals such as Woodstock Hospital, Tillsonburg District Memorial Hospital, and Alexandra Hospital in Ingersoll. Additionally, walk-in clinics, Oxford Settlement

Services, the Muslim Association (mosques and Islamic schools), libraries, churches, and newcomer and refugee support organisations throughout Oxford County. All organisations were within Oxford County's rural boundaries. To ensure a diverse and information-rich sample, both purposive and snowball sampling methods were used. Purposive sampling enabled the intentional selection of individuals with direct experience and knowledge relevant to the research questions. Snowball sampling was then used to facilitate additional recruitment through participant referrals. This strategy is beneficial for reaching harder-to-access groups, as Vincent notes that "snowball sampling is a classic tool used to access 'hard to reach' populations" [54:110]. These methods offered flexibility, efficiency, and cost-effectiveness.

Verbal informed consent was obtained from all participants before each Zoom interview. Individuals who expressed interest after viewing study posters/flyers were contacted by email and provided with the Informed Consent Form (ICF) in advance, outlining the study purpose, procedures, and participant rights. At the start of each Zoom interview, the interviewer (the first author, a doctoral student) reviewed the informed consent form (ICF) with participants and invited questions to confirm understanding. Verbal informed consent was obtained before the interview began and was documented by the student investigator on the study consent record (including the investigator's name, signature, and date). The student investigator then signed the ICF and provided the participant with an electronically signed copy for their records before commencing the interview. Interviews were digitally recorded via Zoom, and recordings, transcripts, and study documents (including consent documentation) were stored on an institutionally secure server with access restricted to the investigators (the first author and the principal investigator), in accordance with the approved institutional ethics protocol. In total, 25 participants (16 men and nine women) were enrolled.

## Data collection

Data collection was conducted through in-depth, semi-structured interviews with open-ended questions. All interviews were conducted in English, which focused on participants' experiences navigating the healthcare system. Although the interview guide was informed by existing literature on healthcare access barriers, the data analysis followed an inductive approach. The themes were not predetermined. Instead, they derived directly from the interview data through an iterative process of coding and comparison. After transcription, the first author conducted open coding of the interviews, allowing patterns and themes to develop from participants' accounts. These themes were subsequently refined through discussion with the co-authors to ensure analytical rigour and coherence.

All interviews were conducted via Zoom using audio recording. Use of the video function was optional and determined by participant preference; some participants chose to enable video, while others preferred to keep it turned off. This approach was adopted to respect participants' comfort and privacy while ensuring interview continuity in the event of internet instability. Each interview was scheduled for up to 60 minutes; the average duration was approximately 45 minutes, with interviews ranging from [minimum] to [maximum] minutes. Participant compensation was not based on the quality of responses, the content of the interview, or any form of performance evaluation. Instead, the modest variation in compensation [20–25 CAD] reflected practical considerations such as interview length and scheduling burden. All participants were informed in advance of the compensation range, participation was entirely voluntary, and compensation was provided regardless of whether participants completed the full interview. The compensation amount and procedure were reviewed and approved by the institutional research ethics board as part of the overall study protocol, prior to data collection. Individual interviews were not reviewed separately; instead, the approved protocol governed all interviews conducted in the study. This approach ensures fair compensation for each participant's contribution. However, some participants declined the token and chose to participate voluntarily.

## Data analysis

The data were examined using reflexive thematic analysis (RTA) following Braun and Clarke [55,56], aligning with the study's constructivist stance. RTA was approached as a fluid, iterative method rather than a rigid sequence of

steps.. The transcripts were read and initially analysed by the first author. This involved repeated reading of the transcripts, reflexive note-taking, and the development of preliminary codes and themes. The emerging analysis was subsequently discussed with the co-authors, who provided critical feedback and guidance to support reflexivity, analytical rigour, and interpretation. Inductive and interpretive coding was conducted by the first author, focusing on generating analytic observations of participants' experiences with rural healthcare rather than on predetermined or line-by-line codes. The coding framework and emerging interpretations were subsequently discussed with the co-author (the principal investigator), who provided critical feedback and contributed to refining themes and interpretations. NVivo 15 assisted with organising the material, but did not drive the analytic decisions. Next, preliminary themes were developed by grouping interconnected codes and considering how they addressed the research questions while situating them within the broader social and structural realities of rural Oxford County. The review and refinement of early themes were conducted by the doctoral candidate investigator (first author). This process involved revisiting the entire dataset to refine, merge, or remove themes, ensuring each had a clear and coherent central concept. The refined themes were subsequently discussed with the co-author (the principal investigator), who provided constructive feedback and contributed to the final interpretation. Afterwards, themes were clearly defined and named, accompanied by concise descriptions and selected data excerpts. Consistent with RTA, themes were viewed as interpretive accounts shaped through the researcher's engagement with the data, not as objective findings that simply "appeared." The first author's subjectivity was treated as an analytic asset and made explicit through ongoing reflexive practice. To enhance transferability, that is, the extent to which qualitative findings may be applicable or relevant to other contexts through rich, detailed descriptions that enable readers to make their own judgments rather than relying on statistical generalisation, the analysis and reporting incorporated detailed descriptions of the study context. This included characterising rural Oxford County as a geographically dispersed, agricultural region shaped by winter conditions and significant travel distances, as well as detailing participants' social positions and healthcare experiences [57,58]. Finally, in the sixth stage, the first author wrote the interpretive narrative and connected it to the RTD framework, demonstrating how structural forces shape and limit access to healthcare.

### Ethics statement

This study received ethics approval from the Hamilton Integrated Research Ethics Board (HiREB #18145) on 10 January 2025. Human participants were prospectively recruited between 10 January 2025 and 30 April 2025. Verbal informed consent was obtained from all participants prior to enrolment. The consent process was approved by HiREB.

## Results

The goal of this study is to explore how ethnic minorities in Oxford County, rural Ontario, perceive their access to healthcare services and to identify the significant barriers that, through the lens of their human rights using the RtD framework, impact both their right to development and the achievement of equitable health outcomes. The qualitative analysis identified two primary themes related to healthcare barriers affecting ethnic minorities in rural Oxford County: 1) constraints of social support, and 2) appraisal of equity. Each theme highlights specific barriers that shape healthcare access and utilisation for these communities. The list of themes and their corresponding subthemes, identified during the analysis, is presented in Table 1. The study identified themes reflecting barriers faced by ethnic minority participants. These included constraints in social support, with sub-themes of ethnoracial stratification, social isolation, and limited social support, as well as themes related to equity and inclusion in policy, including accessibility, resource allocation, and inclusive care.

The section that follows is a detailed account of the significant themes, enriched by selected quotations that capture participants' lived experiences.

**Table 1. List of themes and subthemes.**

| Themes | Subthemes |
|---|---|
| Constraints of Social Support | Ethnoracial Stratification; Social Isolation; and Limited Social Support |
| Equity and Inclusion Policy | Accessibility; Resource Allocation; and Inclusive Care |

## Theme one | constraints of social support

Social support is critical for individual well-being and successful access to care. However, in many rural areas, barriers such as racial segregation, social isolation and barriers to healthcare can leave vulnerable groups isolated and under-served [59]. This theme explores how social dynamics and resource gaps combine to limit meaningful support, deepening health and social inequalities for minorities and those with special needs.

**Subtheme - ethnoracial stratification.** Ethnoracial Stratification refers to the hierarchical ordering of racial and ethnic groups within a society, where access to power, resources, and opportunities is unequally distributed along racial and ethnic lines. This system produces and maintains advantages for dominant groups while disadvantaging minority groups through structural, institutional, and everyday practices.

> EM025 adds: "I noticed that white people are just very separated from the other communities. They do not really, you know, mix in, or they just treat you as a brown person…They are not very friendly; they are just very separate."

> EM012 adds: "I feel like there are obviously some barriers related to healthcare, and like people of colour."

These accounts highlight ethnic segregation as more than a background feature of community life; participants described it as a social context that shaped how they anticipated and experienced healthcare. EM025's description of being treated "as a brown person" indicates a racialised sense of being marked as an outsider, which can produce heightened vigilance and uncertainty when interacting with institutions perceived as predominantly White. In this context, healthcare is not approached as a neutral service but as a setting where unequal treatment is possible and where belonging is not assumed. EM012's comment that barriers exist for "people of colour" reinforces that participants understood these challenges as patterned rather than individual. These narratives suggest that social separation and limited intergroup contact may restrict access to informal support (e.g., guidance on where to seek care, how to navigate services) while also undermining trust and comfort in healthcare encounters. This can contribute to delayed care-seeking, reduced willingness to ask questions or advocate for needs, and diminished continuity of care mechanisms, thereby making community-level segregation a practical barrier to equitable healthcare access. This statement highlights the ongoing challenges that ethnic minority communities face in accessing equitable healthcare. The participant highlights the ongoing challenges that ethnic minority communities face in accessing equitable healthcare.

> EM016 traced their sense of social division back to their school years, where rigid groupings by skin colour first took root: "There would be like specific groups of like one color of skin, another group of people that are like one colour of skin. There were not that many people who had the same group of likes."

A recurring theme in participants' accounts was the sense of exclusion or differential treatment based on visible markers of identity.

> EM007 said: "From the so, the average they are treated Muslims differently. I do not know why?"

This finding reflects participants' experiences of perceived religious discrimination that was felt but not always easily articulated or attributed to a single cause. EM007's uncertainty, "I do not know why", highlights how differential treatment may be experienced as subtle, normalised, or challenging to name, particularly when it intersects with racialised and cultural identities. Such ambiguity can make discrimination harder to challenge or report, while still shaping feelings of exclusion and vulnerability in healthcare settings. This illustrates how intersecting racial, cultural, and religious identities can compound barriers to equitable healthcare access, even when discriminatory practices are not overt or explicitly acknowledged.

**Subtheme - social isolation.** Segregation and stigma result in minority individuals feeling isolated, fearful of judgment, and reluctant to seek help. Participants described day-to-day encounters that made them feel excluded or unwelcome. Even basic gestures, like offering a greeting, were often disregarded.

EM019 said: "If I say Hi to them all morning, they will just ignore me… I think I am the only Black. If I am not mistaken."

This quote illustrates everyday racialised exclusion that contributed to feelings of isolation and shaped expectations of exclusion in healthcare settings. Experiences like this reinforce feelings of social isolation and being overlooked within the community. When such treatment is repeated, it can undermine individuals' sense of belonging and make it harder for minority group members to participate fully in community life.

EM0022 adds: "My friend, an immigrant woman, told me that when she walks, she notices people looking at her and talking, just because of how they see her."

EM001 participants also described how nonverbal cues conveyed disapproval more powerfully than words: "But some of the people were very quiet, and they did not say anything. And their faces could tell that they are feeling that."

She adds: "my son, he was not able to make any friends like for a couple of years. He did not have any friend or like, you know, someone who was very close to him or he was not able to like, you know, mix in."

These narratives demonstrate that social isolation is not limited to individual adults but can have a profound impact on entire families, including children. Participants described broader social changes in their communities, such as increased cultural diversity and perceived declines in social cohesion, which they felt had altered everyday interactions and a sense of belonging.

EM004 stated: "And part of this is also a breakdown of the social structure. There is no family structure anymore in most places in Canada. Parents are living on their grand…. So that's a social structure, deterioration problem as well."

This explanation reflects participants' perceptions of a weakening of family and community support structures in the broader Canadian context. Such informal networks were described as important sources of emotional and practical support, particularly for navigating healthcare systems when faced with language barriers, unfamiliarity with services, or caregiving responsibilities. Participants understood the erosion of these support structures as contributing to increased isolation and vulnerability, thereby indirectly shaping barriers to healthcare access. In the absence of family or community support, individuals were less able to seek guidance, accompaniment, or advocacy, which discouraged help-seeking and limited engagement with healthcare services.

**Subtheme - limited support systems.** Barriers to healthcare access are multifaceted, affecting minority groups at multiple stages, during care, after discharge, and in educational support for children. Participants identified practical obstacles that hinder their ability to access support and essential services, particularly for those who are new to the community or speak languages other than the local one.

EM004 said: "There is an issue of translation sometimes. So, language barriers…. In Woodstock, there are not many people who speak Hindi, Arabic, or other languages besides English and potentially French. So, the language barrier is one piece. The other barrier in Oxford County is the distance."

These reflections underscore the significant barriers to accessing healthcare and other services that linguistic isolation creates.

EM025 participants highlighted insufficient resources in the school system for children with disabilities, which creates further obstacles to timely and effective care: "The school for children with special needs does not have teachers for disabled kids or special needs kids. They do not have funding for the teachers. The class sizes are increasing."

These findings highlight the pressures placed on hospital and frontline staff working within resource-constrained rural systems, where limited staffing and funding increase workloads and heighten the risk of burnout. Participants' accounts also point to fragmented service provision, underscoring the need for more integrated social, educational, and healthcare services to support children with disabilities and their families in rural communities.

EM004 emphasised the unexpected in formal care once patients leave the hospital: "The patient will come to the hospital…. and stay for 2, 3, 4 days. They are stabilised to some degree, then discharged. Once they are discharged, there is no proper place for them to go. There is typically no job for them to do, so they end up in the same situation, either in home shelters or on the street, with a similar lifestyle. And then if they were into drugs, they get back into the drugs."

This highlights a critical post-discharge support gap; without stable housing, employment or addiction services, patients are left vulnerable to homelessness and deterioration.

**Theme two | equity and inclusion policy**

The theme of equity appraisal reflects participants' calls for practical policy changes to address persistent inequities in healthcare delivery, particularly for ethnic minority communities in rural areas. Their insights can be grouped into three key areas: accessibility, resource allocation, and inclusive care. Each subtheme highlights specific barriers and actionable recommendations to make the healthcare system fairer, more responsive, and culturally competent.

**Subtheme - accessibility.** A key concern raised by participants is the importance of making healthcare services genuinely accessible for everyone, particularly ethnic minorities and those in rural communities. The availability of culturally responsive healthcare was identified as a consistent theme.

EM021 underline: "Make it more accessible for the doctors, professionals who are immigrants who are already here, to make it easy for them to be in a position to serve people. So, people have more access to healthcare, and immigrant doctors can help immigrant minorities with language barriers and everything. They will be more considerate… they understand where they are coming from."

Participant highlights the expectation that living in a developed country ensures access to free, high-quality healthcare. However, newcomers often discover that accessibility goes beyond cost; it also involves timely, effective, and adequate services.

EM007 said: "Health is related to development. Yes, because we live in a developed country, right? So, when we were coming to Canada, I thought the medical system was perfect. But when I came here, I saw, oh my God! There is a terrible situation. Of course, these are free. Of course, these are free, but if anything is free, you know, we cannot compromise on the quality."

Their experience underscores that true accessibility in healthcare means not just removing financial barriers but also guaranteeing quality and effectiveness for everyone. Participants also emphasised the critical need for improved infrastructure and resource allocation.

> EM017 suggested practical solutions: "In rural sectors where hospitals and clinics may be few and far between, robust community health programs, such as mobile clinics and telehealth services, can significantly improve quality of life."

> EM013 stressed: "Offering more language translation services." EM014 recommended broader patient advocacy initiatives: "Creating a patient advocacy program to help navigate the healthcare system and establishing partnerships with local organizations to tailor healthcare services to cultural needs. These initiatives aim to enhance accessibility and ensure equitable healthcare opportunities."

These quotes highlight a gap between participants' expectations of Canada's healthcare system and their lived experiences. While Canada was perceived as a developed country with a high-quality healthcare system, participants reported frustration with long wait times, delayed access to services, and concerns about compromised quality of care, even within a publicly funded system. Additionally, participants emphasised structural barriers, particularly language barriers and difficulties navigating the healthcare system, and called for enhanced translation services and patient advocacy programs. Overall, the narratives point to a need for more accessible, culturally responsive, and patient-centred healthcare services to ensure equitable care. The participant highlights the need for patient advocacy, navigation support, and community partnerships to help individuals better understand and access healthcare services. It also emphasises the importance of culturally responsive care to reduce barriers, improve accessibility, and promote equitable healthcare opportunities.

> EM014: "Creating a patient advocacy program to help navigate the healthcare system and establishing partnerships with local organizations to tailor healthcare services to cultural needs. These initiatives aim to enhance accessibility and ensure equitable healthcare opportunities."

Such efforts help bridge gaps for individuals who might otherwise struggle to access or benefit from healthcare, making the system more inclusive and responsive to community needs. These suggestions underscore the importance of addressing physical, linguistic, and systemic barriers in creating a healthcare environment where all individuals can easily access the care they need.

**Subtheme - resource allocation.** Participants emphasised the necessity for fair and strategic distribution of resources, from funding to personnel, to strengthen healthcare systems in underserved areas. Participants highlighted that funding and resource availability were significant barriers to progress.

> EM001 noted: "There should be more funding given to the healthcare department... because I think there is a lot of shortage. The government needs to spend a lot more on this aspect rather than other things because it is very important."

> EM014 highlighted this perspective, suggesting targeted financial policies: "Even if that means safeguard putting tax up, which is always a controversial idea, but maybe we need to increase taxes by half a penny, half a cent or one cent, or two cents, so that we have got better funding for long-term healthcare because we have an aging population."

Participants expressed concerns about ongoing underfunding in the healthcare system and emphasised the need for increased government investment. Despite recognising that taxes are already substantial, some participants indicated a willingness to accept small, incremental tax increases if these funds were directed toward improving healthcare services. This reflects a view of healthcare as a public priority and an understanding of the financial pressures created by workforce

shortages and the needs of an ageing population. At the same time, these perspectives highlight a tension between the existing tax burden and the perceived necessity of sustainable, long-term funding to maintain and enhance healthcare quality. Participants also underscore the necessity for fair and strategic distribution of resources, from funding to personnel, to strengthen healthcare systems in underserved areas.

> EM014 further elaborated: "We need to encourage those students in rural areas to go to school and become healthcare professionals and encourage them in terms of jobs so that they do not all go to the big cities, that they stay around in rural areas like Woodstock."

> EM015 addressed this issue: "The second option is to take in more foreign-trained doctors. There are doctors who are coming in, but they are either getting discouraged from writing the exams or taking the exams, or even if they took the exams, not finding employment."

> EM010 reinforced this sentiment: "I think if the government makes it a policy that makes it at least easier, not as extreme as what it is now, there will be more doctors around the world and in Oxford County."

Focusing on workforce policies, both cultivating local talent and easing the path for experienced international professionals, resource allocation reforms can help close persistent gaps in care, especially for rural and minority populations.

**Subtheme - inclusive care.** Finally, participants consistently advocated for care that respects cultural backgrounds and individual needs through specialised training and tailored services.

> EM015 highlighted the necessity for culturally sensitive healthcare policies and practices: "To improve these experiences, it is crucial for healthcare policies and practices to be culturally sensitive and inclusive, ensuring that all community members feel respected and adequately cared for. There is a critical need for healthcare systems to train providers in cultural competence, enabling them to recognise and appropriately respond to these cultural factors. This training helps build trust and improve communication between providers and patients, ensuring that medical advice is both culturally appropriate and effective."

> Similarly, EM010 advocated for comprehensive policy reforms: "Improving equity often requires policy reforms that mandate cultural competence training, expanded interpreter services, and better infrastructure to address the unique healthcare challenges of rural communities."

While these measures are designed to promote equity, they also directly advance inclusivity by ensuring that healthcare services are accessible and responsive to the needs of diverse populations.

> EM023 clearly expressed this concern: "Please increase the standard of education of your doctors before you place them in the emergency room (ERs). ER is a very specialised branch and should be better than general practitioners. Doctors should be more qualified than a GP."

The quote does not merely express concern about training but also reveals assumptions about professional status and legitimacy. Suggesting that ER doctors should be "more qualified than a GP," the participant implies a hierarchical distinction between emergency medicine and general practice, positioning the former as superior. Such assumptions implicitly devalue general practitioners and frame them as insufficiently competent to be included in emergency care settings. In this way, appeals to "standards" of education function as barriers to inclusivity by reinforcing exclusionary professional boundaries and undermining multidisciplinary practice. Spiritual and emotional support in healthcare settings was also identified as a significant aspect requiring attention.

EM008 emphasised: "More work can be done to have spiritual support. For instance, in my case, someone who understands Islam can be close to them. The majority of the doctors do not really have any theological beliefs at all."

Embedding cultural competence, respect, and individualised support within healthcare delivery is central to achieving equity and ensuring that everyone feels understood and valued by the system. Together, these recommendations demonstrate that achieving equitable healthcare requires comprehensive reforms. Improving accessibility, ensuring fair resource allocation, and embedding inclusive, culturally sensitive practices are essential steps toward a system where everyone, regardless of background or location, can access high-quality care and feel respected throughout their healthcare journey.

## Discussion

The first theme centres on ethno-racial stratification, social isolation, and gaps in social support. In Oxford County, limited interaction between the predominantly white, conservative Christian majority and ethnic minority communities reflects national patterns in which minorities are portrayed as isolated or difficult to integrate [59,60]. These dynamics mirror broader sociological findings on how racial boundaries are reproduced through institutions and daily interactions. The separation of white residents from ethnic minorities, along with differential treatment based on visible identity, aligns with theories of the "Self" and "Other," where inclusion and exclusion shape group identities [51]. Such boundaries are maintained through attitudes, everyday practices, and subtle social cues that reinforce mistrust and limit meaningful contact.

Ideas of "civilisation," "savagery," and inherent group differences were embedded in Enlightenment philosophy [61]. Philosophers such as Locke and Voltaire contributed to the naturalisation of racial hierarchies, enabling forms of dehumanisation and discrimination that have influenced Western institutions more broadly [51]. In the Canadian context, these intellectual traditions have shaped colonial governance, immigration policy, and institutional norms, including within healthcare systems [62,63].Within a RtD framework, equitable access to healthcare requires attention not only to material resources but also to the social conditions that shape inclusion and participation. In Oxford County, patterns of social separation and differential treatment reported by participants indicate that racialised boundaries continue to influence healthcare experiences. While existing literature (e.g., discussions of "Self" and "Other" or Orientalism) helps contextualise these dynamics, the RtD framework highlights that such exclusion directly undermines individuals' ability to participate in and benefit from healthcare systems. Addressing these barriers is therefore essential to fulfilling RtD principles of equity, inclusion, and non-discrimination [63].

These legacies are relevant to the present study, as they provide essential context for understanding how racialised and Muslim patients may experience exclusion, stereotyping, or diminished legitimacy within healthcare settings. Addressing such inequities, therefore, requires not only policy change but engagement with the deeper cultural and social boundaries that persist within Canadian society.

Building on this regional distinction, Kilgour [63] argues that historical traditions play an important role in shaping ethnic identity formation. At the same time, Satzewich and Liodakis emphasise that present-day treatment by outsiders further influences how ethnic groups understand their social position and belonging [51,64]. Identity is therefore fluid, shaped by ancestry and ongoing social experience. While participants did not state this explicitly, their accounts can be understood, when situated within existing scholarship, as reflecting broader historical patterns in which Western societies have positioned themselves as morally superior to non-Western cultures [61,65]. Cultural relativism challenges this by asserting the equal worth of all cultures [66, 67], while multiculturalism supports coexistence within legal and ethical limits. Although Canada's multicultural policies promote immigrant integration [67,68], tensions persist, as shown by rising hostility toward Muslim women after the 2015 Paris attacks [15]. This debate also reflects the clash between cultural relativism and universalism; the latter is embedded in the UDHR [33] and often criticised as Western-centric. Still, rights related to basic needs and health have proven beneficial in liberal democracies [69], provided they are implemented with political commitment and cultural awareness [70]. From a researcher's standpoint, these tensions emphasise the need to trace

the distance between policy ideals and everyday social realities. Examining how individuals negotiate identity, belonging, and recognition in local contexts reveals the gap between formal frameworks and lived experience, highlighting the structural and interpersonal forces that shape inclusion and exclusion. In Oxford County, everyday distancing behaviours, hesitant interactions, differential greetings, and segregated gathering spaces mirror the "Self/Other" framework and reveal emotional discomfort with cultural differences; these behaviours reinforce minority isolation. Participants described being ignored, judged through nonverbal cues, or treated as outsiders. Children experienced social exclusion in schools. These patterns reflect research showing that race-related stress harms health regardless of socioeconomic status [71]. National data also indicate persistent racial disparities in primary health indicators.

The sense of being singled out deepens vulnerability, consistent with Rohwerder and Szyp's [72] findings. Loneliness is widespread in Canada, particularly among immigrant seniors [73], and is associated with poorer health and reduced social participation [74]. These findings indicate the need for gender- and migration-sensitive policies addressing isolation. This focus is necessary because isolation does not affect all groups equally; men and women experience distinct social expectations, caregiving roles, and access to support networks, while migrants encounter additional challenges related to language, cultural adjustment, and disrupted family structures. These intersecting factors shape vulnerability in different ways and require policies that respond to the specific conditions faced by each group.

Sociological theories situate these experiences within broader patterns of disconnection. - Weakened social ties limit individuals' ability to participate fully in society, a concern central to the RtD framework. This aligns with Émile Durkheim's concept of anomie, which describes the breakdown of social bonds and collective norms [75]. Similarly, Karl Marx's notion of alienation points to a related but distinct form of disconnection one rooted in economic structures, where individuals are separated from the products of their labour, from their own productive activity, and from their human essence. Both anomie and alienation, despite their different origins, identify conditions that undermine the social and economic foundations necessary for meaningful participation in development [76]. These patterns reflect broader processes of social exclusion that can limit participation and access to resources, which are central concerns within the RtD framework. The sub-theme on limited social support shows that barriers extend far beyond securing appointments. Language is a significant obstacle; the lack of interpreters for languages like Hindi and Arabic restricts healthcare access and civic participation [77,78]. Because language is tied to identity, limited communication undermines belonging and the exercise of rights [79]. Critics argue that multicultural and bilingual policies often obscure deeper power inequalities [80,81]. With immigrants comprising 23% of Canada's population [1], addressing linguistic and cultural barriers is increasingly urgent. Language barriers and digital literacy gaps have hindered access to public health information, contributing to poorer health outcomes among newcomers [82,83]. Evidence supports professional interpretation, culturally responsive care, and newcomer involvement in healthcare design [84]. These patterns suggest that meaningful progress requires more than surface-level language accommodations; it calls for sustained engagement with the cultural, institutional, and relational factors that shape communication, belonging, and trust in rural healthcare settings.

Participants also described gaps in community supports following hospital discharge, including unstable housing, unemployment, and limited addiction services, which contribute to repeated crises. These cycles strain healthcare workers and lead to burnout [20,85]. Hospitals, designed for acute care, become responsible for unmet social needs [86,87]. These deficits reflect broader social conditions: limited networks heighten distress and weaken coping capacities, while isolation contributes to chronic illness and mortality [88]. Children with special needs face additional inequities due to staffing shortages and funding constraints in rural schools [89,90].

These patterns show that social support gaps in Oxford County reflect more profound structural inequalities affecting immigrants, minorities, and rural populations. Moving beyond symbolic multiculturalism toward socially inclusive development is essential [91]. A rights-based framework, which is part of RtD, highlights the universality of rights, including healthcare, housing, and social services [33,92], and emphasises the obligation to deliver services as close as possible to communities [93]. Rights are interdependent, as highlighted by the 1986 UN Declaration on the Right to Development

[25], and meaningful development requires participation, transparency, and non-discrimination [26]. The findings demonstrate that Oxford County's social support deficits are symptoms of more profound structural inequalities. Addressing them requires culturally informed interventions, stronger community infrastructure, and policy approaches guided by the RtD, ensuring solutions that reflect local realities while upholding universal rights.

The findings from the equity and inclusion policy theme show that healthcare accessibility extends beyond affordability and service availability. It requires timely, high-quality, culturally responsive care. Participants noted that although Canada is widely viewed as offering universal, free healthcare, many newcomers and minority residents experience significant gaps in service quality and effectiveness. These gaps are apparent in rural communities with limited infrastructure and uneven service delivery. Steinfeld and Maisel [94] emphasise that participatory design is essential for achieving genuine universality and accessibility, as involving diverse users ensures that systems reflect a wide range of needs. Daniels [95] similarly argues that inclusive decision-making transforms abstract commitments to universality into concrete, equitable outcomes. Without such inclusive processes, systems presented as universal or accessible can continue to reproduce hidden forms of exclusion.

Universal design and design for all aim to ensure that public systems can be used by everyone, regardless of background or ability [95]. These approaches align closely with social justice principles and are embedded in major international frameworks, such as the UN Convention on the Rights of Persons with Disabilities, which positions accessibility and universality as interconnected rights that enable full participation in society. As Persson and colleagues [5] note, the two concepts form the practical foundation for equitable participation and outcomes. This study, therefore, places universality and accessibility at the centre of healthcare system design, underscoring the need for policies that respond to the lived realities of ethnic minorities in rural areas.

A key insight from this theme is the importance of culturally responsive care, including supporting immigrant doctors and health professionals who share patients' cultural backgrounds. Such practitioners help build trust, increase comfort, and reduce misunderstandings that hinder access. Drawing on Max Weber's analysis of social action and authority, cultural characteristics can also shape what [96] calls "monopolistic closure," a process in which dominant groups use shared traits to secure exclusive access to valued resources, thereby excluding others. This form of exclusion is precisely what the Rtd framework seeks to dismantle. Cultural responsiveness in healthcare is therefore essential because when systems fail to accommodate diverse backgrounds, they reproduce the very exclusion that the RtD framework seeks to address. The RtD requires that states ensure equal access to essential resources, including healthcare, without discrimination a principle consistent with Article 8 of the 1986 Declaration on the Right to Development, which guarantees equality of opportunity in access to health services and other basic resources which has identified in Art 8 of UN General Assembly [25].

This exclusion becomes reinforced through institutions and everyday practices [97]. Weber further notes that social boundaries often rest on minor or imagined differences that gain significance through collective belief [96]. Movements of people through migration, colonisation, or labour bring these distinctions into sharper focus. It is therefore not the objective weight of cultural differences but their social construction that enables monopolistic closure and sustains boundaries.

Operationalising these ideas, accessibility in health equity refers to "the extent to which systems, environments, and services can be used by people with the widest possible range of characteristics and capabilities" [95]. When treated as a universal value, accessibility becomes a foundation for universality, ensuring that everyone can meaningfully participate in and benefit from health and social systems. This requires more than the availability of services; it demands usable, inclusive services free from physical, linguistic, social, and cultural barriers [98]. Article 8 of the Declaration on the RtD states that states must ensure equal access to essential resources, including healthcare [25]. This obligation encompasses equity, participation, and non-discrimination, ensuring that no group is systematically excluded.

Access to healthcare is essential for improving health and preventing illness [99]. In high-income countries with basic services, the discussion shifts to service quality, fairness in distribution, and the timeliness and effectiveness of care. The

WHO 2008 identifies health as a fundamental human right and asserts that universal access is indispensable for realising this principle. When access is inadequate, use of services declines and health outcomes worsen [100,101]. Although Canada's healthcare system is formally universal [102], significant disparities persist across populations and regions [39]. Access to healthcare is widely regarded as a basic social entitlement, rooted in the belief that everyone deserves care regardless of whether providing it yields economic benefits [103]. These insights reinforce the idea that universality in name does not guarantee universality in practice; meaningful access depends on systems that recognise the diverse circumstances shaping people's ability to seek, navigate, and benefit from care.

Inclusive development has long served as a central principle of global human rights and social policy. The International Covenant on Economic, Social and Cultural Rights, adopted in 1966, established key protections, while later conventions, such as the Convention on the Rights of the Child in 1989 and the Convention on the Rights of Persons with Disabilities in 2006, expanded guidance for specific populations [104]. The 2030 Sustainable Development Goals reinforce these commitments; Target 10.2 calls for full participation and empowerment of all individuals, irrespective of identity or status [40]. Inclusive development sits at the core of the RtD, which envisions that all people can participate in, contribute to, and benefit from development [104]. The UNDP similarly emphasises that development becomes inclusive only when all groups contribute, benefit, and participate in decision-making, grounded in the principles of participation, non-discrimination, and accountability [98].

These insights align with the first author's perspective, a doctoral candidate, that meaningful inclusion requires dismantling systemic barriers rather than relying on symbolic diversity measures. From this perspective, inclusive development depends on both removing obstacles and creating genuine avenues for participation among disadvantaged groups. While inclusion is essential in health policy, it alone cannot determine how resources are allocated. Rawls [24] argues that justice demands prioritising the needs of the least advantaged, even when these diverge from majority preferences. The Oregon Experiment, a landmark U.S. healthcare rationing initiative from the late 1980s that used public consultation to rank and prioritise medical services for public funding, illustrates that while transparency and public input are important, fairness ultimately depends on the meaningful inclusion of disadvantaged groups [104]. Daniels and Sabin [20] further stress that just health policy requires transparency, reasoned justification, mechanisms for appeal, and accountability. Accordingly, health policy must prioritise accessibility and universality, ensuring that ethnic minorities and other marginalised populations can obtain necessary care. This approach moves beyond procedural inclusion to address substantive requirements, making healthcare meaningfully universal and equitable in practice.

## Limitations of the study

This study offers important insights into healthcare barriers for ethnic minorities in rural Oxford County, but several limitations exist. The qualitative approach and purposive sampling restrict the ability to generalise findings. The findings may be influenced by self-selection bias, and while objectivity was prioritised, interpretation ultimately rests with the researcher. The study reflects conditions at a specific time and place, and changes in policy or demographics could affect future relevance. The first author's social positioning, including being an Arabic minority researcher and not residing in the rural community studied, shaped the interpretive lens through which the analysis was conducted. In line with principles of reflexive thematic analysis, the researcher's subjectivity was understood as an inevitable and productive element of knowledge production rather than a source of bias to be eliminated. Reflexivity was therefore treated as an ongoing, analytic practice, involving continuous engagement with how assumptions, experiences, and interpretive decisions informed coding, theme development, and meaning-making throughout the analytic process. This approach emphasised transparency and critical self-awareness in the construction of themes, rather than positioning positionality as a fixed characteristic disclosed only at the outset of the research. Some participants hesitated to share openly due to concerns about confidentiality, which may have limited the depth of responses. Future research should include larger, more diverse samples from multiple regions and provide multilingual interview options to ensure greater inclusivity and trust.

## Conclusion

This study explores healthcare access barriers faced by ethnic minority populations in rural Oxford County, Ontario, through the lens of the RtD. As discussed in the introduction, there is growing international recognition that development must be participatory, inclusive, and grounded in human rights rather than solely in economic or institutional progress. The findings of this study contribute to this emerging discourse by illustrating how, in practice, limited social support and culturally insensitive care can prevent rural ethnic minority populations from fully participating in and benefiting from health and social systems, even where formal rights and services exist. While inclusion is widely promoted as a core value within contemporary policy frameworks, the findings underscore that inclusion alone is insufficient. Policies must also guarantee universality, ensuring equal claims to services, and address practical accessibility by tackling the real-world barriers that persist despite formal entitlements. This aligns with the "basic needs" approach to development, which developed from recognition that economic growth alone cannot eliminate poverty or secure well-being, and echoes Rawls' concept of primary social goods. Both perspectives emphasise that just societies and effective development policies must prioritise the fundamental needs of the most disadvantaged, rather than focusing solely on aggregate progress or symbolic participation.

Accordingly, this study suggests that healthcare reform in rural Ontario must integrate procedural values, such as meaningful participation, with substantive commitments to universality and accessibility. Addressing intersecting barriers requires moving beyond top-down policies to collaborative action, ensuring all individuals, especially minorities, have both the right and the practical ability to access high-quality, culturally appropriate care. Advancing health equity is a matter of social justice and an essential element of sustainable development, requiring coordinated policy action, cross-sector collaboration, and ongoing accountability. Setting ambitious goals, ensuring transparency, and involving those most affected can make universality and diverse participation a reality, particularly in rural and minority communities where exclusion is embedded in everyday experiences. Embedding these human rights principles into policy brings us closer to a development model where everyone benefits.

## Acknowledgments

My thanks go to the Mary Heersink School of Global Health at McMaster University for supporting my PhD research. I also appreciate the Hamilton Integrated Research Ethics Board for evaluating my study protocol and providing helpful suggestions. I am especially grateful to my committee members, particularly Professor Sara Bannerman, for their consistent support and insightful feedback.

## Author contributions

**Conceptualization:** Amal Jawad.

**Data curation:** Amal Jawad.

**Formal analysis:** Amal Jawad.

**Funding acquisition:** Bonny Ibhawoh.

**Investigation:** Amal Jawad.

**Methodology:** Amal Jawad, Bonny Ibhawoh, Lisa Schwartz, Andrew Kapoor.

**Project administration:** Amal Jawad, Bonny Ibhawoh, Lisa Schwartz, Andrew Kapoor.

**Resources:** Amal Jawad.

**Software:** Amal Jawad.

**Supervision:** Lisa Schwartz, Andrew Kapoor.

**Validation:** Amal Jawad, Bonny Ibhawoh.

**Visualization:** Amal Jawad, Andrew Kapoor.

**Writing – original draft:** Amal Jawad.

**Writing – review & editing:** Amal Jawad, Bonny Ibhawoh, Lisa Schwartz, Andrew Kapoor.

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
