## [Decision Letter · Decision Letter 0]

26 Jan 2026

PGPH-D-25-03893

The Right to Development and Disparities in Healthcare Access: Qualitative Study from Rural Ontario, Canada

Dear Dr. Jawad,

Thank you for submitting your manuscript to PLOS Global Public Health. After careful consideration, we feel that it has merit but does not fully meet PLOS Global Public Health’s publication criteria as it currently stands. Therefore, we invite you to submit a revised version of the manuscript that addresses the points raised during the review process.

A letter that responds to each point raised by the editor and reviewer(s). You should upload this letter as a separate file labeled ’Response to Reviewers’.A marked-up copy of your manuscript that highlights changes made to the original version. You should upload this as a separate file labeled ’Revised Manuscript with Track Changes’.An unmarked version of your revised paper without tracked changes. You should upload this as a separate file labeled ’Manuscript’.

We look forward to receiving your revised manuscript.

Kind regards,

Tsitsi B. Masvawure, Ph.D.

Academic Editor

Journal Requirements:

Additional Editor Comments (if provided):

Dear Authors,

I invite you to respond to the reviewers’ comments. Reviewer 1 provided extensive feedback in the body of your paper, so pease review these in detail. In addition to the reviewers’ comments, please consider condensing the Intro, Theoretical framework and Discussion sections. These are currently too long and could be reduced in length. Furthermore, consider engaging with some of the participant excerpts you share in the findings/results section. On page 13, for instance, you present five participant excerpts but do not unpack them in any detail. Consider choosing fewer excerpts which you then explain and unpack for the reader in detail. Participant excerpts can be interpreted in multiple ways and so it is your job as the authors to tell the readers how you are interpreting participant data. Additionally, please engage explicitly with the concept of RtD in the Discussion, since this is the theoretical underpinning for the study.

Sincerely,

Tsitsi B Masvawure

Reviewers’ comments:

Reviewer’s Responses to Questions

**Comments to the Author**

1. Does this manuscript meet PLOS Global Public Health’s publication criteria? Is the manuscript technically sound, and do the data support the conclusions? The manuscript must describe methodologically and ethically rigorous research with conclusions that are appropriately drawn based on the data presented.

Reviewer #1: Yes

Reviewer #2: Yes

2. Has the statistical analysis been performed appropriately and rigorously?

Reviewer #1: N/A

Reviewer #2: Yes

3. Have the authors made all data underlying the findings in their manuscript fully available (please refer to the Data Availability Statement at the start of the manuscript PDF file)?

Reviewer #1: No

Reviewer #2: Yes

4. Is the manuscript presented in an intelligible fashion and written in standard English?

Reviewer #1: Yes

Reviewer #2: Yes

5. Review Comments to the Author

Reviewer #1: This paper presents a worthwhile discussion of health equity from the context of rural Canada, from the perspective of the RtD framework, which is of increasing interest globally. I thank the authors for their submission, which I genuinely appreciated reading.

I congratulate the authors on the introduction and discussion sections, which were generally well-written. However, the results section would need a significant amount of work. As it stands, the analysis is far too thin, and at some points unclear. I would suggest taking a closer look at how others in global health and social medicine have analysed and reported qualitative data. In terms of approach: it is unclear how the paper operationalised Braun & Clarke’s thematic analysis approach -- at some points, there were contradictions that indicate areas of misunderstanding (I have noted some of these sections, and provided some resources that might help). There are also some typos and other areas of unclear text, which I have noted in my comments, which are all available in the attached document.

I hope you’d find my comments constructive, and wish you all my very best.

Reviewer #2: This is amazing work that appeals to the rights of residents of Canada especially minorities in remote communities. Not only is the work you did very applaudable, but it also justly fills the gap many EDI contexts lack: "the voice of the affected". A grammatical suggestion, however, for this article is to include in brackets, RtD, the abbreviation for Right to Development, at its first mention. Great work once again!

6. PLOS authors have the option to publish the peer review history of their article (what does this mean?). If published, this will include your full peer review and any attached files.

**Do you want your identity to be public for this peer review?** For information about this choice, including consent withdrawal, please see our Privacy Policy.

Reviewer #1: No

Reviewer #2: No

Figure Resubmissions:

---

## [Decision Letter · Decision Letter 1]

23 Apr 2026

PGPH-D-25-03893R1

The Right to Development and Disparities in Healthcare Access: Qualitative Study from Rural Ontario, Canada

Dear Dr. Jawad,

Thank you for submitting your manuscript to PLOS Global Public Health. After careful consideration, we feel that it has merit but does not fully meet PLOS Global Public Health’s publication criteria as it currently stands. Therefore, we invite you to submit a revised version of the manuscript that addresses the points raised during the review process.

Please submit your revised manuscript by . If you will need more time than this to complete your revisions, please reply to this message or contact the journal office at globalpubhealth@plos.org. When you’re ready to submit your revision, log on to https://www.editorialmanager.com/pgph/ and select the ’Submissions Needing Revision’ folder to locate your manuscript file.

A letter that responds to each point raised by the editor and reviewer(s). You should upload this letter as a separate file labeled ’Response to Reviewers’.A marked-up copy of your manuscript that highlights changes made to the original version. You should upload this as a separate file labeled ’Revised Manuscript with Track Changes’.An unmarked version of your revised paper without tracked changes. You should upload this as a separate file labeled ’Manuscript’.

We look forward to receiving your revised manuscript.

Kind regards,

Tsitsi B Masvawure, Ph.D.

Academic Editor

Journal Requirements:

Additional Editor Comments (if provided):

A reviewer raised additional concerns based on the revisions you made to your paper. I therefore invite you to consider the following in your next revision:

a) Please respond to the reviewer’s comment about using the word "emerge/emerging" when referring to your thematic analysis. If you agree with their suggestion to refrain from using these terms then please review the paper thoroughly and change this language. If you disagree with the reviewer’s suggestion, please provide an explanation in your response letter.

b) I concur with the reviewer that your discussion often ends up straying from the RtD framework when you reference  philosophers, Said and other theories etc. Please revise this section so that it remains focused on RtD and does not introduce too many other theories.

Thank you for your responsiveness to the reviewers’ comments.

Reviewers’ comments:

Figure Resubmissions:

---

## [Editor Report · Decision Letter 2]

18 May 2026

The Right to Development and Disparities in Healthcare Access: Qualitative Study from Rural Ontario, Canada

PGPH-D-25-03893R2

Dear Ms Jawad,

We are pleased to inform you that your manuscript ’The Right to Development and Disparities in Healthcare Access: Qualitative Study from Rural Ontario, Canada’ has been provisionally accepted for publication in PLOS Global Public Health.

If your institution or institutions have a press office, please notify them about your upcoming paper to help maximize its impact. If they’ll be preparing press materials, please inform our press team as soon as possible -- no later than 48 hours after receiving the formal acceptance. Your manuscript will remain under strict press embargo until 2 pm Eastern Time on the date of publication. For more information, please contact globalpubhealth@plos.org.

Best regards,

Tsitsi B Masvawure, Ph.D.

Academic Editor

Thank you for your latest revisions. The Discussion now reads a lot more cohesively.